# Paraventricular Mast Cell-Derived Histamine Activates CRH Neurons to Mediate Adult Visceral Hypersensitivity Induced by Neonatal Maternal Separation

**DOI:** 10.3390/brainsci13111595

**Published:** 2023-11-16

**Authors:** Ziyang Chen, Tiantian Zhou, Yunfan Li, Tingting Li, Zhengnian Ding, Li Liu

**Affiliations:** 1Department of Anesthesiology, The First Affiliated Hospital of Nanjing Medical University, Nanjing 210029, China; 2Department of Anesthesiology, Nanjing Integrated Traditional Chinese and Western Medicine Hospital Affiliate with Nanjing University of Chinese Medicine, Nanjing 210014, China; 3Department of Geriatrics, Jiangsu Provincial Key Laboratory of Geriatrics, The First Affiliated Hospital of Nanjing Medical University, Nanjing 210029, China; ltt@njmu.edu.cn

**Keywords:** early-life stress, visceral hypersensitivity, mast cells, CRH neurons, histamine

## Abstract

Neonatal maternal separation (NMS) is an early-life stress (ELS) that can result in adult visceral hypersensitivity, which is usually manifested as chronic visceral pain. Although mast cells and corticotropin-releasing hormone (CRH) neurons are involved in stress response, whether there is an interaction between mast cells and CRH neurons in hypothalamic paraventricular nucleus (PVN) during the ELS-induced visceral hypersensitivity remains elusive. Herein, we established an NMS model by separating neonatal mice from their mothers, and observed that these mice presented visceral hypersensitivity in adulthood, as indicated by elevated abdominal withdrawal reflex and lowered visceral pain threshold. The NMS-induced adult visceral hypersensitivity was accompanied by activation of mast cells and CRH neurons in PVN. Also, NMS increased the histamine content (an inflammatory mediator mainly released by mast cells) and histamine H2 receptor (H2R) expression of CRH neurons in PVN. Remarkably, intra-PVN administration with mast cell stabilizer attenuated the NMS-induced CRH neuronal activation and adult visceral pain, while histamine administration showed the opposite effects. Moreover, intra-PVN injection with H2R antagonist alleviated the NMS-induced CRH neuronal activation, PKA and CREB phosphorylation, and importantly, adult visceral pain. Together, our findings revealed a role of an interaction between paraventricular mast cells and CRH neurons in NMS-induced adult visceral hypersensitivity, thereby providing a perspective for the management of visceral pain.

## 1. Introduction

Visceral hypersensitivity, mainly manifested by chronic visceral pain, is an important clinical feature of functional gastrointestinal diseases such as irritable bowel syndrome. With a high clinical incidence, visceral hypersensitivity seriously affects patients’ life quality and consumes a large amount of medical resources [1,2]. The etiology of visceral hypersensitivity is complicated and may involve physical and/or psychological stresses and other factors such as genetic and environmental triggers [3,4]. Notably, people subjected to stress, especially early-life stress (ELS), exhibit increased vulnerability to visceral pain disorders [5]. Neonatal maternal separation (NMS) is a typical ELS known to trigger visceral pain behavior [6,7,8]. However, the mechanisms underlying the lasting impact of ELS on the development of adult visceral hypersensitivity remains largely unknown.

ELS has a great effect on the developmental trajectory of the central nervous system (CNS), particularly hypothalamic–pituitary–adrenal axis, rendering individuals more susceptible to the onset of chronic pain in adulthood [5,9]. As a vital component of the hypothalamic–pituitary–adrenal axis, the hypothalamic paraventricular nucleus (PVN) holds a pivotal function in the modulation of ELS-induced chronic visceral pain [10,11]. Corticotropin-releasing hormone (CRH) neurons are important cells mediating the stress response and are widely distributed in the PVN [12,13]. Our previous studies have indicated the significance of CRH neurons in the PVN in regulating the chronic visceral hypersensitivity triggered by ELS [14]. However, the precise mechanism responsible for paraventricular CRH neuronal activation in the development of ELS-induced visceral hypersensitivity remains elusive.

Previous studies have shown that neuroimmune interactions within CNS play a crucial role in mediating the change in the neural function and behavior induced by stress [15]. Recent evidence has suggested that mast cells, a type of immune cell of the CNS, are significantly implicated in the progression of chronic pain [16,17]. Mast cells act not only as primary responders during harmful events, but also as environmental “sensors” that interact with neurons [18,19]. In this pathological state, activated mast cells discharge numerous proinflammatory mediators, including histamine, which amplify the perception of pain by heightening neuronal excitability to facilitate synaptic transmission [20,21]. We recently demonstrated that NMS activates paraventricular mast cells to induce visceral hypersensitivity in rats [22]. However, it is unknown whether there is an interaction between mast cells and CRH neurons during visceral hypersensitivity induced by ELS.

In this study, using an NMS-induced adult visceral hypersensitivity model in mice, we revealed that paraventricular mast cells release histamine to activate CRH neurons, which mediates visceral hypersensitivity. The results indicate that blocking the communication between mast cells and CRH neurons could be considered an alternative therapeutic strategy for the management of visceral hypersensitivity.

## 2. Methods and Materials

### 2.1. Animals

Preweaning male C57BL/6 mice were purchased from the Laboratory Animal Center of Nanjing Medical University and housed in standard plexiglass cages under constant temperature and humidity (22 °C and 50%). The animals were kept under a 12 h light and 12 h dark cycle, with ad libitum access to chow and water. All experimental procedures were performed following the guidelines set by the International Association for the Study of Pain and were approved by the Institutional Animal Care and Use Committee at Nanjing Medical University.

### 2.2. Reagents

The reagents were as follows: Cromolyn and histamine (Sigma, St. Louis, MO, USA); Ranitidine (MedChemExpress, Monmouth Junction, NJ, USA); Avidin Alexa Fluor^TM^488 (Thermo Fisher, Waltham, MA, USA); Sheep polyclonal anti-CRH antibody (Novus, Littleton, Colorado, USA); Rabbit anti-c-Fos antibody (Cell Signaling Technology, Boston, MA, USA); Rabbit polyclonal anti-H1R antibody (Alomone Labs, Jerusalem, Israel); Rabbit polyclonal anti-H2R antibody (Alomone Labs, Jerusalem, Israel); Rabbit polyclonal anti-H3R antibody (Alomone Labs, Jerusalem, Israel); Rabbit anti-phospho-PKA (T197) antibody (Abcam, Cambridge, UK); Rabbit anti-phospho-CREB (S133) antibody (Abcam, Cambridge, UK); Mouse anti-GAPDH antibody (Bioworld Technology, St. Louis Park, MN, USA); Mouse histamine ELISA kit (ColorfulGene Biological Technology, Wuhan, China); Alexa Fluor^TM^488 donkey anti-sheep IgG (Life Technologies, Carlsbad, CA, USA); Alexa Fluor^TM^594 donkey anti-rabbit IgG (Abcam, Cambridge, UK).

### 2.3. NMS Induction

Neonatal maternal separation (NMS) was conducted to induce visceral hypersensitivity as previously described [22]. Briefly, the pups in the NMS group were separated from their mothers for 6 h each day (8:00–11:00 a.m. and 14:00–17:00 p.m.) from postnatal day 2 to day 15. During the separation periods, all pups from each litter were taken out of their home cage and placed into an incubator to keep them warm. Following this separation, the pups were returned to their dams. The control pups and their mothers were housed together without any handling. On postnatal day 21, the pups were weaned and separated from their mothers. The tests were conducted when the mice reached 8 weeks of age.

### 2.4. Assessment of Visceral Sensitivity

Visceral sensitivity was determined by abdominal withdrawal reflex (AWR) scores and visceral pain threshold. As described by a previous study [8], the mice were lightly anesthetized with sevoflurane and a homemade latex balloon, with dimensions of 2 cm in length and 1.3–1.5 cm in diameter, which was fastened to a catheter and linked to a sphygmomanometer and syringe using a three-way pipe. The balloon was carefully inserted into the colorectum 0.5 cm above the anus to induce colorectal distention. AWR scores were recorded in response to different colorectal pressures (20, 40, 60, or 80 mmHg), each applied for 20 s, with a 4 min break. AWR scoring criteria: 0, no significant behavioral changes; 1, immobility or head movement; 2, light contraction of abdominal muscles, but no lifting from the desktop; 3, clear contraction of abdominal wall muscles and lifting from the desktop; 4, arching of the abdominal wall and back. The visceral pain threshold was defined by the noticeable contraction of the abdominal wall. Each distension procedure was performed in triplicate and the results were averaged for subsequent analysis.

### 2.5. Intra-PVN Microinfusion

Mice were anesthetized with sevoflurane and placed in a stereotaxic apparatus (RWD Life Science, Shenzhen, China). After making an incision in the skin to expose the skull, a stainless steel needle was inserted into the PVN. Then, we determined the precise coordinates of the PVN region in accordance with Paxinos and Watson brain atlas: AP: −0.94 mm, ML: ±0.25 mm, DV: −5.00 mm. To examine the effects of cromolyn (mast cell stabilizer, 10 μg in 0.2 μL), histamine (4 μg in 0.2 μL), and ranitidine (H2R antagonist, 10 μg in 0.2 μL) on the visceral hypersensitivity, the reagents were administered into the PVN of mice via a microinjection pump for more than 3 min. The injection needle remained in place for 10 min to facilitate complete diffusion of the solution, and then the pain threshold was tested.

### 2.6. Western Blot Analysis

Following anesthesia, mice were swiftly sacrificed. The brain tissues encompassing the PVN region were carefully excised, placed on ice, and subsequently stored at −80 °C.

Then, the samples were rapidly placed into a RIPA lysis buffer with PMSF. Subsequently, they underwent homogenization and were centrifuged at 12,000 rpm for 15 min at 4 °C. Equal quantities of protein were subjected to electrophoresis using SDS-PAGE gels and transferred onto the PVDF membranes. After blocking with 5% non-fat milk for 2 h at room temperature on an agitator, the PVDF membranes were incubated with anti-CRH (1:500), anti-H1R (1:500), anti-H2R (1:500), anti-H3R (1:500), anti-phospho-PKA (1:500), anti-phospho-CREB (1:500), and anti-GAPDH (1:1000) primary antibody overnight at 4 °C. The PVDF membranes were left to incubate with the HRP-conjugated secondary antibody (1:1000) at room temperature for a duration of 2 h. Protein bands were observed utilizing an enhanced chemiluminescence kit.

### 2.7. Immunofluorescence Staining

Mice were deeply anesthetized and transcardially perfused with 0.9% saline, followed by 4% polyformaldehyde. The entire brains, which included the PVN, were promptly removed and then fixed in 4% polyformaldehyde at 4 °C for 24 h, followed by immersion in 30% sucrose. The brain tissues were coronally sliced at a thickness of 20 μm using a cryostat (Leica Biosystems, Heidelberg, Germany). Brain slices were blocked with 10% donkey serum at room temperature for 2 h, followed by incubation with anti-CRH (1:200), anti-c-Fos (1:200), and anti-H2R (1:200) primary antibody at 4 °C for 24 h. Immunofluorescence staining of mast cells was performed using avidin Alexa Fluor^TM^488 (1:200). Subsequently, the slices were rinsed three times and incubated with a fluorescent secondary antibody at room temperature for 2 h. Finally, the tissue sections were mounted on positively charged slides and visualized using fluorescence microscopy (Olympus, Tokyo, Japan).

### 2.8. Mast Cell Staining by Toluidine Blue

Brain tissue sections encompassing the PVN region were prepared, stained with a 0.1% solution of toluidine blue, and subjected to cell counting according to the established protocols [17]. These slides were submerged in the staining solution for 30 min, followed by two rinses with distilled water. Subsequently, they underwent dehydration using a sequence of escalating ethanol concentrations, and were finally submerged in butyl acetate ester. The slides were allowed to dry overnight. Mast cells were counted with Cell D 1.15 software (Olympus, Tokyo, Japan).

### 2.9. Enzyme-Linked Immunosorbent Assay (ELISA)

The level of histamine in the PVN was measured using ELISA kits. Briefly, the PVN was extracted and homogenized using a RIPA lysis buffer. The homogenates were centrifuged at 12,000 rpm for 20 min at 4 °C. The supernatants were collected and repeatedly analyzed using histamine ELISA kits according to the manufacturer’s instructions.

### 2.10. Statistical Analysis

All data are presented as the mean ± SEM. Differences between two groups were assessed using Student’s two-tailed unpaired *t*-test. For comparison of multiple groups, we employed one-way ANOVA or two-way repeated measures ANOVA followed by Bonferroni’s post hoc test. All data were analyzed using GraphPad Prism 9.0 software (GraphPad, San Diego, CA, USA). A value of *p* < 0.05 was considered statistically significant.

## 3. Results

### 3.1. NMS-Induced Visceral Hypersensitivity in Adulthood Was Accompanied with Mast Cell Activation in PVN

To establish an ELS-induced adult visceral hypersensitivity, NMS was performed in neonatal mice between postnatal day 2 and 15. The adult mice (8 weeks of age) that underwent NMS exhibited increased AWR scores following administration with colorectal distention pressures at 40, 60, and 80 mmHg, respectively, compared to the distention pressure-matched non-NMS control mice (two-way repeated measures ANOVA test, Treatment: F (1,5) = 56.25, *p* = 0.0007; Pressure: F (3,15) = 176.2, *p* < 0.0001; Treatment × Pressure: F (3,15) = 1.793, *p* = 0.1918; Figure 1A). In contrast, the adult mice that underwent NMS exhibited reduced visceral pain threshold compared to the non-NMS control mice (t (8) = 5.96, *p* = 0.0003, Figure 1B).

Next, we evaluated the mast cell abundance and activation in PVN. Toluidine blue staining presented that NMS increased the mast cell number in PVN of adult mice compared with control mice (t (4) = 7.363, *p* = 0.0018, Figure 1C). Also, fluorescence staining of avidin showed more mast cellular granules in PVN of NMS-treated adult mice compared with the control adult mice without NMS treatment (t (4) = 6.8888, *p* = 0.0023, Figure 1D).

### 3.2. Pharmacological Inhibition of Mast Cells Alleviates NMS-Induced Adult Visceral Hypersensitivity

To examine the role of mast cell activation in NMS-induced adult visceral hypersensitivity, a mast cell stabilizer cromolyn was administered to adult mice that underwent NMS. Cromolyn decreased the mast cell number in PVN of NMS-treated adult mice (t (4) = 7.071, *p* = 0.0021, Figure 2A). Moreover, cromolyn decreased the mast cellular granules in PVN of NMS-treated adult mice (t (4) = 6.217, *p* = 0.0034, Figure 2B). Following inhibition of mast cell activation with cromolyn in NMS-treated adult mice, the visceral pain threshold was significantly improved from 0.5 h to 2 h after cromolyn injection, with a peak effect at 1 h following cromolyn injection (two-way repeated measures ANOVA test, Treatment: F (1,5) = 345.4, *p* < 0.0001; Time: F (4,20) = 8.749, *p* = 0.0003; Treatment × Time: F (4,20) = 13.94, *p* < 0.0001, Figure 2C).

### 3.3. Inhibition of Mast Cells Alleviates the NMS-Induced CRH Neuronal Activation in PVN

Given that the activation of paraventricular CRH neurons mediates NMS-induced visceral pain, we examined whether CRH neurons can be activated by mast cells following NMS. To this aim, the mast cell stabilizer cromolyn was administered to the NMS-treated adult mice. As shown in Figure 3A, NMS upregulated CRH protein expression in PVN of adult mice, and this upregulation of CRH was abolished following cromolyn treatment (one-way ANOVA test, F (3,8) = 62.99, *p* < 0.0001). Immunofluorescence analysis revealed an elevation in the proportion of c-Fos-positive CRH neurons in PVN of NMS-treated adult mice compared with controls, while this increase was abrogated following cromolyn treatment (one-way ANOVA test, F (3,8) = 57.84, *p* < 0.0001, Figure 3B).

### 3.4. Mast Cell-Derived Histamine Activates CRH Neurons in PVN to Mediate Visceral Hypersensitivity

Mast cells activation can release a wide variety of pro-inflammatory mediators, with histamine being an important inflammatory mediator produced by activated mast cells [21]. Therefore, we examined whether mast cells regulate visceral hypersensitivity via histamine. ELISA results demonstrated that NMS increased histamine levels in PVN of adult mice (t (4) = 11.68, *p* = 0.0003, Figure 4A), and this increase was removed following treatment with mast cell stabilizer cromolyn (t (4) = 8.649, *p* = 0.0010, Figure 4B). Remarkably, histamine administration induced a significant reduction in the visceral pain threshold in normal control mice (two-way repeated measures ANOVA test, Treatment: F (1,5) = 58.95, *p* = 0.0006; Time: F (4,20) = 3.406, *p* = 0.028; Treatment × Time: F (4,20) = 13.94, *p* = 0.0015, Figure 4C).

To determine the direct role of histamine in CRH neuronal activation and visceral hypersensitivity, normal control mice were intra-PVN injected with exogenous histamine. As shown in Figure 4D, histamine administration increased CRH protein expression in PVN compared with saline-treated controls (t (4) = 6.59, *p* = 0.0027). Meanwhile, the proportion of c-Fos-positive CRH neurons was also increased following histamine treatment (t (4) = 13.73, *p* = 0.0002, Figure 4E).

### 3.5. Histamine H2 Receptor (H2R) Mediates NMS-Induced Visceral Hypersensitivity

Histamine exerts its effects through histamine receptors on effector cells. Histamine receptors in the hypothalamus mainly include H1R, H2R, and H3R [23]. To clarify which types of histamine receptors are involved in the histamine-induced CRH neuronal activation, we first examined the expression of H1R, H2R, and H3R in the PVN. Western blot analysis exhibited increased H2R protein expression in PVN of NMS-treated adult mice (t (4) = 5.614, *p* = 0.0049), whereas the protein expression of H1R (t (4) = 0.1447, *p* = 0.8919) and H3R (t (4) = 0.2929, *p* = 0.7841) remained unchanged (Figure 5A). Moreover, the H2R expression was located in CRH neurons in PVN (Figure 5B).

To investigate the involvement of paraventricular H2R in NMS-induced CRH neuronal activation and visceral hypersensitivity, NMS-treated adult mice were intra-PVN administered with H2R antagonist ranitidine. As shown in Figure 5C, NMS-treated adult mice displayed elevated pain thresholds after ranitidine administration for 0.5, 1, and 2 h (two-way repeated measures ANOVA test, Treatment: F (1,5) = 69.62, *p* = 0.0004; Time: F (4,20) = 9.092, *p* = 0.0002; Treatment × Time: F (4,20) = 8.826, *p* = 0.0003).

### 3.6. H2R Mediates the NMS-Induced Activation of PKA–CREB Signaling

Previous studies have shown that histamine binding to H2R affects neuronal excitability mainly through the activation of PKA–CREB signaling pathway [24]. Therefore, we examined whether PKA–CREB signaling was involved in H2R-mediated CRH neuronal activation following NMS. As shown in Figure 6, NMS upregulated phosphorylation levels of PKA (one-way ANOVA test, F (3,8) = 46.70, *p* < 0.0001) and CREB (one-way ANOVA test, F (3,8) = 37.61, *p* < 0.0001) and protein expression of CRH (one-way ANOVA test, F (3,8) = 57.05, *p* < 0.0001) in PVN of adult mice; however, these increases induced by NMS were abrogated by intra-PVN administration with H2R antagonist ranitidine.

## 4. Discussion

In this study, we explored the interaction between paraventricular mast cells and CRH neurons during NMS-induced visceral hypersensitivity. We showed that NMS activates mast cells to release histamine, which activates CRH neurons through H2R, and ultimately leads to visceral hypersensitivity in adulthood (Figure 7).

ELS is a major risk factor for the onset of chronic visceral pain [25]. Early life, especially the neonatal period, is critical for the development of the CNS, is sensitive to environmental changes, and susceptible to permanent structural or functional changes from internal or external environments [26,27]. NMS is a well-characterized ELS and has been widely used as an experimental stress to mimic chronic visceral pain [28,29]. In this study, NMS-treated mice developed an increased AWR score and decreased visceral pain threshold in adulthood, suggesting that NMS indeed resulted in adult visceral hypersensitivity.

Patients with stress-associated visceral pain showed aberrations in brain activity, as evidenced by functional MRI studies [30]. The hypothalamus PVN is a key brain region that mediates the stress response and is implicated in modulating pain behavior [31,32]. Our previous work demonstrated that the hypothalamic PVN region is a crucial regulator of chronic visceral pain induced by ELS [14]. Therefore, in this study, we focused on PVN to explore the cellular and molecular mechanisms associated with chronic visceral pain and found an important role of the interaction between mast cells and CRH neurons in the development of NMS-induced adult visceral hypersensitivity.

Neuroinflammation in the CNS has been identified as the primary factor contributing to chronic pain [33,34]. Mast cells, being components of the innate immune system, represent the “first responders” to various stress conditions rather than microglia [35]. Activated mast cells exert deleterious effects as the primary pathogenic element. A recent study shows that brain mast cell activation plays a central role in neuropathic pain, depression, and neurodegenerative disease [19,36]. In the current study, we found that NMS induced visceral hypersensitivity in adult mice and enhanced mast cells activation in the PVN, while intra-PVN administration with mast cell stabilizer alleviated visceral hypersensitivity evoked by NMS. Our data revealed that paraventricular mast cell activation mediates NMS-induced visceral hypersensitivity.

Previous studies have highlighted the significance of mast cell–neuron interaction in the development of chronic pain, encompassing visceral pain [37,38]. In the mammalian CNS, paraventricular CRH neurons serve as crucial integration centers, regulating the neuroendocrine reaction to stress [13]. CRH is an important hormone mediating the stress response and is involved in visceral hypersensitivity by acting on CRH receptors. Studies have shown that the CRH1 and CRH2 receptors are extensively expressed among immune cells in the intestines [39]. CRH has the potential to activate intestinal immune cells and trigger the release of large amounts of inflammatory mediators. These proinflammatory agents sensitize peripheral nociceptors and heighten visceral perception, leading to the development of visceral pain [5]. Our previous research showed that ELS-induced visceral hypersensitivity is accompanied by activation of CRH neurons in the PVN [14]; however, whether mast cells mediate the NMS-induced paraventricular CRH activation is unknown. In the present study, we found that both mast cells and CRH neurons were activated in PVN following NMS, while inhibition of mast cell activation abrogated the NMS-induced CRH neuronal activation. The data indicate that paraventricular CRH neurons can be activated by mast cells following NMS in mice.

As the “first responder” cells of the CNS, mast cells are capable of sensing environmental stimuli and releasing numerous pro-inflammatory mediators, including histamine [40]. Histamine can activate adjacent sensory neurons and the nerve conduction pathway, potentially contributing to visceral pain development [21,41]. Inhibition of mast cells to release histamine could reduce neuronal damage [42]. However, whether mast cell-derived histamine modulates CRH neuronal activation is unknown. In the current study, we found that NMS increased histamine levels in PVN, and intra-PVN injection with histamine activated CRH neurons and sensitized mice to colorectal distention-induced visceral pain. Moreover, when the mast cell stabilizer increased visceral pain threshold in adult NMS-treated mice, it decreased histamine levels in PVN. Together, the results suggest that histamine mediates CRH neuronal activation by mast cells following NMS.

Histamine receptors (HR) mediate the activation of histamine. Histamine receptors in the hypothalamus mainly include H1R, H2R, and H3R [23]. In this study, NMS increased only H2R expression but not H1R and H3R expression in PVN of adult mice, and H2R expression was present in CRH neurons. Following inhibition of H2R, the NMS-induced visceral hypersensitivity in adult mice was also alleviated. Considering that histamine binding to H2R activates adenylate cyclase to generate cAMP for activating protein kinase A (PKA) and CREB, this drives gene expression to activate the neurons [24,43,44]. Intriguingly, we found in this study that NMS increased CRH protein expression and increased phosphorylation levels of PKA and CREB, while H2R antagonist diminished these increases induced by NMS. The findings suggest that PKA–CREB signaling might be involved in H2R-mediated CRH neuronal activation. Importantly, the experiments conducted in this research were exclusive to male mice, and it should be noted that the findings from male subjects might not completely translate to female mice. Prior research has indicated the significant impact of ovarian hormones on visceral hypersensitivity [45,46,47]. Thus, male mice were specifically chosen for this study to minimize the influence of estrogen on visceral pain. Due to the heightened intensity of visceral pain in female rodents and humans, it is imperative that future research prioritizes female mice and examines the impact of sex differences on visceral pain.

## 5. Conclusions

In conclusion, we defined an effect of paraventricular mast cells on CRH neuronal activation for the development of NMS-induced visceral hypersensitivity. This action of mast cells was mediated by increased histamine release, which activates H2R-medaited PKA–CREB signaling to activate CRH neurons, and ultimately leads to visceral hypersensitivity. Thus, blocking the interaction between mast cells and CRH neurons may serve as an alternative therapeutic approach for the management of ELS-induced visceral hypersensitivity. Our present findings reveal the potential underlying mechanism contributing to chronic visceral pain and provide new targets for the management of this condition.

## Figures and Tables

**Figure 1 brainsci-13-01595-f001:**
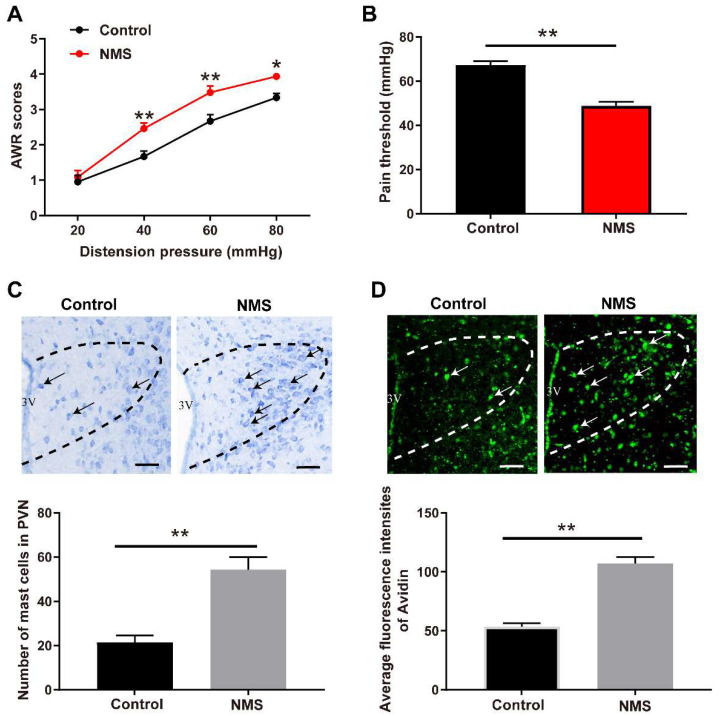
NMS-induced visceral hypersensitivity in adulthood was accompanied with mast cell activation in PVN. (**A**) NMS mice presented increased AWR scores compared with the control mice (two-way repeated measures ANOVA followed by Bonferroni’s post hoc test: * *p* < 0.05, ** *p* < 0.01 vs. Control group, *n* = 6). (**B**) NMS mice displayed a lower visceral pain threshold compared with the control mice (** *p* < 0.01 vs. Control group, *n* = 6). (**C**) Typical images of mast cells in PVN stained with toluidine blue (PVN region: outlined by black dashed lines. Black arrows indicate activated mast cells). Scale bar = 50 μm. NMS mice exhibited a higher mast cell count in PVN compared with the control mice (** *p* < 0.01 vs. Control group, *n* = 3). (**D**) Representative images of PVN mast cells stained with avidin staining (PVN region: outlined by white dashed line. White arrows indicate activated mast cells). Scale bar = 50 μm. NMS mice presented more mast cellular granules in PVN compared with the control mice. (** *p* < 0.01 vs. Control group, *n* = 3).

**Figure 2 brainsci-13-01595-f002:**
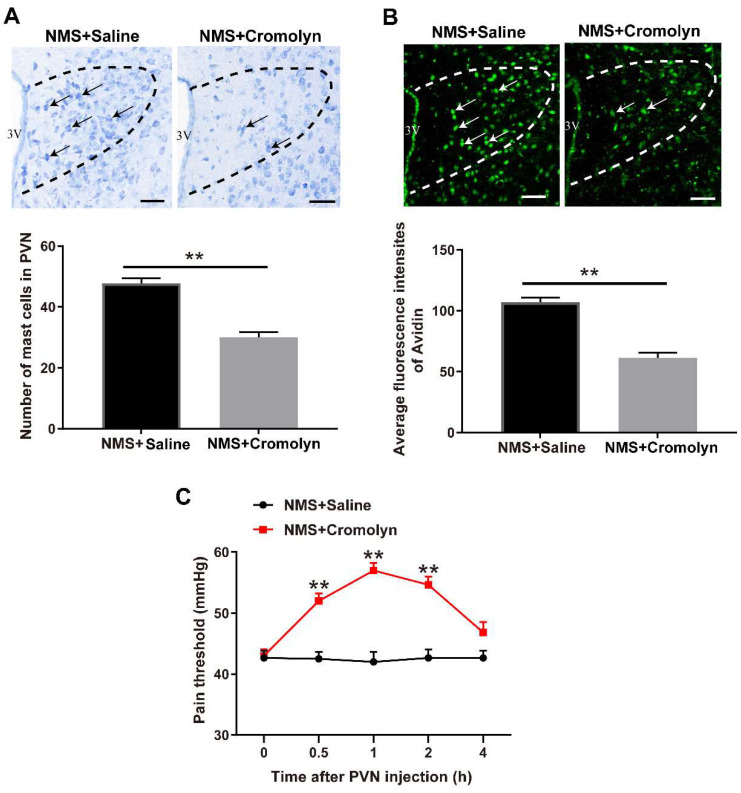
Inhibition of mast cells activation attenuates NMS-induced visceral hypersensitivity. (**A**,**B**) Typical images of PVN mast cell stained with toluidine blue and fluorescence staining of avidin (PVN region: outlined by black and white dashed lines. Black and white arrows indicate activated mast cells). Scale bar = 50 μm. Intra-PVN injection of cromolyn suppressed the mast cells activation induced by NMS (** *p* < 0.01 vs. NMS + Saline group, *n* = 3). (**C**) The time course for mast cell stabilizer cromolyn (10 μg) administration and behavioral test in mice. Intra-PVN infusion of cromolyn prevented an increase in the visceral pain threshold for 0.5, 1, and 2 h in mice that experienced NMS (two-way repeated measures ANOVA followed by Bonferroni’s post hoc test: ** *p* < 0.01 vs. NMS + Saline group, *n* = 6).

**Figure 3 brainsci-13-01595-f003:**
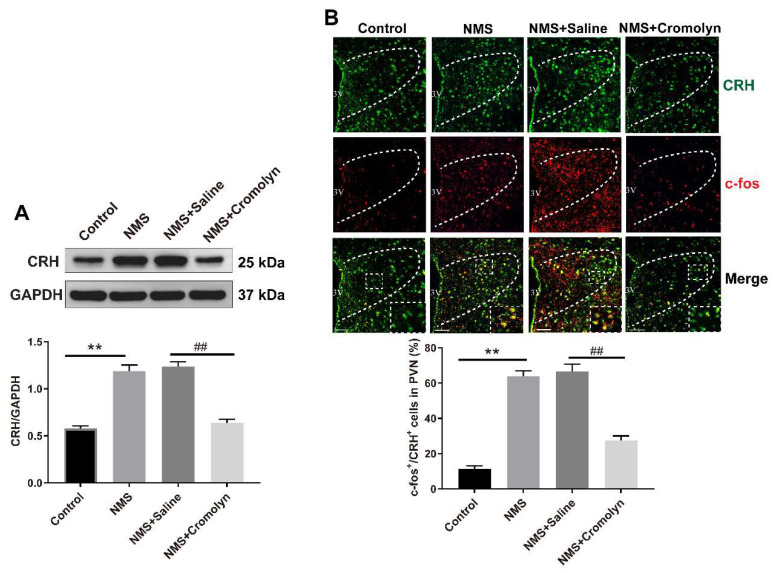
Inhibition of mast cells activation in PVN suppresses NMS-induced CRH neuronal activation. (**A**) NMS mice exhibited increased CRH protein levels in PVN compared with control mice, and injection of cromolyn into PVN prevented the CRH protein elevation (one-way ANOVA followed by Bonferroni’s post hoc test: ** *p* < 0.01 vs. Control group, *^##^ p <* 0.01 vs. NMS + Saline group, *n* = 3). (**B**) Typical images of immunofluorescence labeling for CRH (green) and c-Fos (red) in PVN (PVN region: outlined by white dashed lines). Scale bar = 50 μm. Immunofluorescence analysis exhibited a significant upregulation in the proportion of c-Fos-positive CRH neurons in the NMS group mice compared with the controls, while this increase was prevented following cromolyn treatment (one-way ANOVA followed by Bonferroni’s post hoc test: ** *p* < 0.01 vs. Control group, *^##^ p <* 0.01 vs. NMS + Saline group, *n* = 3).

**Figure 4 brainsci-13-01595-f004:**
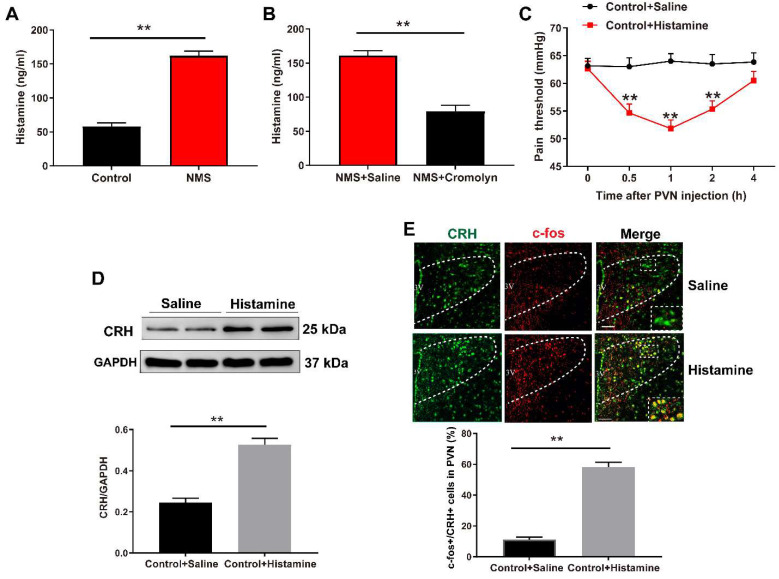
Mast cell-derived histamine activates CRH neurons in PVN to mediate visceral hypersensitivity. (**A**) ELISA was used to detect the histamine levels. NMS mice exhibited an increase in histamine levels in PVN compared with control mice (** *p* < 0.01 vs. Control group, *n* = 3). (**B**) Intra-PVN infusion of cromolyn suppressed the histamine levels (** *p* < 0.01 vs. NMS + Saline group, *n* = 3). (**C**) The timeline for the administration of histamine (4 μg) and the subsequent behavioral test in mice. Exogenous histamine administration in PVN resulted in a decreased visceral pain threshold at 0.5, 1, and 2 h in control mice (two-way repeated measures ANOVA followed by Bonferroni’s post hoc test: ** *p* < 0.01 vs. Control + Saline group, *n* = 6). (**D**) The CRH protein expression in PVN was upregulated after intra-PVN infusion of histamine in control mice (** *p* < 0.01 vs. Control + Saline group, *n* = 3). (**E**) Typical images of immunofluorescence labeling for CRH (green) and c-Fos (red) in PVN (PVN region: outlined by white dashed lines). Scale bar = 50 μm. Exogenous injection of histamine into PVN increased the proportion of c-Fos-positive CRH neurons in control mice (** *p* < 0.01 vs. Control + Saline group, *n* = 3).

**Figure 5 brainsci-13-01595-f005:**
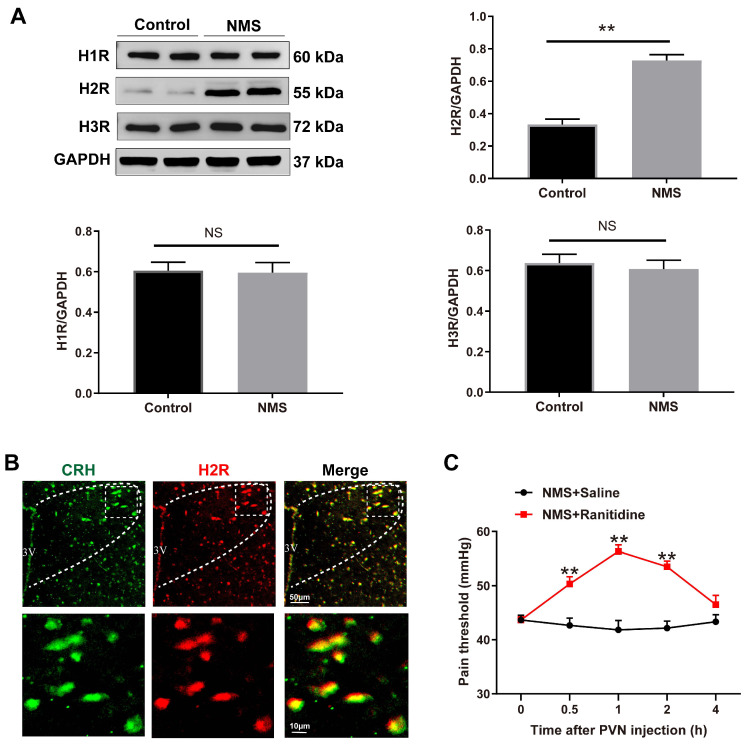
Histamine H2 receptor (H2R) mediates NMS-induced visceral hypersensitivity. (**A**) The expression of H1R, H2R, and H3R in PVN was examined using Western blot analysis. NMS mice displayed an upregulation of H2R protein levels in PVN compared with control mice, whereas the H1R and H3R protein levels were not significantly altered (** *p* < 0.01 vs. Control group, *n* = 3). NS indicate no significance. (**B**) Double-immunofluorescence labeling showed the co-localization of H2R (red) with CRH neurons (green) in PVN (PVN region: outlined by white dashed lines. Yellow color indicate co-localization between H2R and CRH neurons). Scale bar = 50 μm (top); Scale bar = 10 μm (bottom). (**C**) The time course for H2R antagonist ranitidine (10 μg) administration and behavioral test in mice. Intra-PVN infusion of H2R antagonist ranitidine exhibited an increase in the visceral pain threshold for 0.5, 1, and 2 h in NMS mice (two-way repeated measures ANOVA followed by Bonferroni’s post hoc test: ** *p* < 0.01 vs. NMS + Saline group, *n* = 6).

**Figure 6 brainsci-13-01595-f006:**
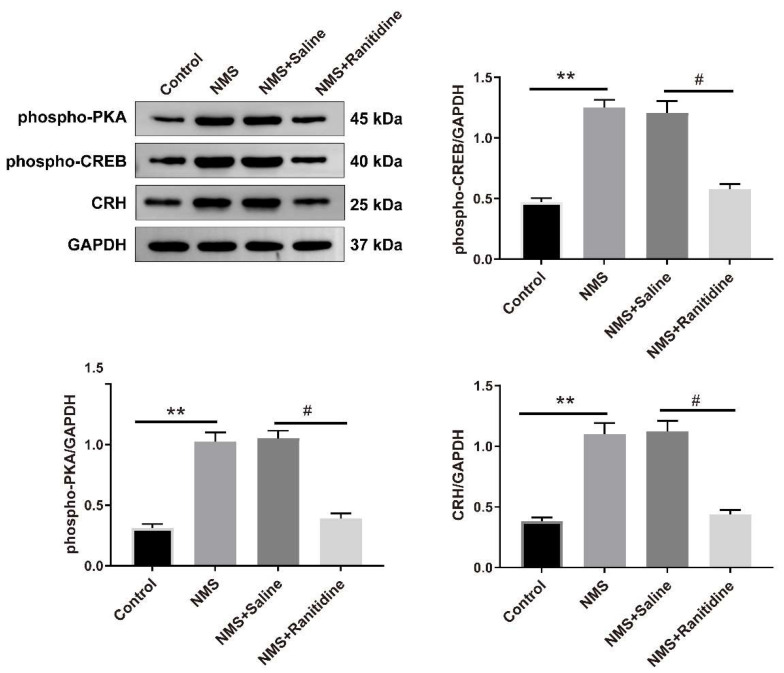
H2R mediates the NMS-induced activation of PKA–CREB signaling. The expression of phosphorylation levels PKA and CREB and CRH protein expression in PVN was detected by Western blot analysis. NMS mice presented an increase in phosphorylation levels of PKA and CREB and CRH protein levels in PVN compared with control mice, while this increase was prevented following H2R antagonist ranitidine treatment (one-way ANOVA followed by Bonferroni’s post hoc test: ** *p* < 0.01 vs. Control group, *^#^ p <* 0.05 vs. NMS + Saline group, *n* = 3).

**Figure 7 brainsci-13-01595-f007:**
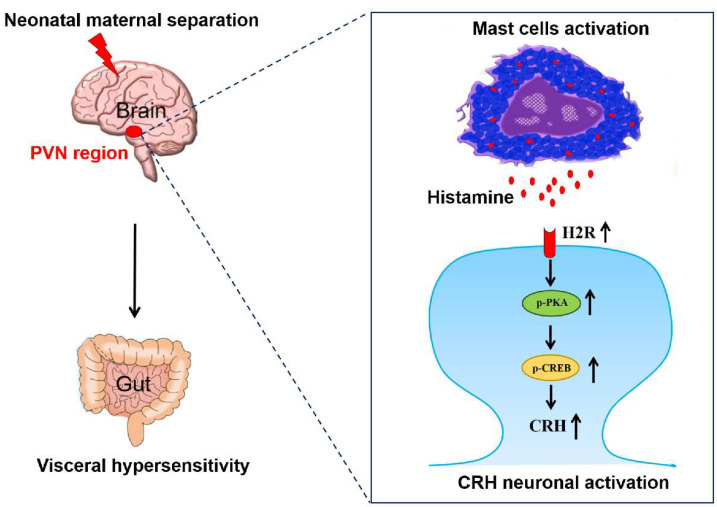
Mechanistic scheme. NMS activates mast cells to release histamine, which activates H2R-medaited PKA–CREB signaling to activate CRH neurons in PVN, and ultimately leads to visceral hypersensitivity in adulthood.

## Data Availability

The data presented in this study are available on request from the corresponding author.

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
