# Peer review of "Paraventricular Mast Cell-Derived Histamine Activates CRH Neurons to Mediate Adult Visceral Hypersensitivity Induced by Neonatal Maternal Separation"

_brainsci, 2023, doi:10.3390/brainsci13111595_

Round 1

Reviewer 1 Report

Comments and Suggestions for Authors

This manuscript describes a pathway by which paraventricular mast cells activate a circuit that leads to chronic visceral pain in an early life stress maternal separation model.

I have a number of issues that need to be addressed.

1. Only males were used in the study. This is a problem. Stress related visceral pain is worse in females and might be modulated by a different mechanism. There is no justification for using only males. Excluding females just to avoid estrous cycle effects is not adequate justification. See DOI 10.1186/s13293-016-0087-5

2. A lot more information is needed to explain the 6 hr NMS per day. Most studies use a 3 hr separation. How were the pups monitored over the 6 hr period? How were they kept warm? How was food and water provided? Pups need to feed every 3-4 hrs, 6 hrs is excessive.

3. during the abdominal withdrawal reflex testing, how were the animals restrained? Was the observer blinded to treatment? For score 3, it states “obvious contraction.” What does that mean?

4. More detail is needed for the PVN microinjections. Was a guide cannula used? What was the duration of the injection, .2 ul injected over how many minutes? How do you know you were in the correct spot?

5. In the results, 3.1: fluorescence staining showed mast cellular granules? What is that? Is it degranulated mast cells? In figure 1, C,D it states 3 animals per group but how many sections were analyzed per animal? Was it done blinded?

6. All the statistical analyses are incomplete. The multiple comparisons are shown in the graphs, but the statistical test used, the overall statistical result, the F and P values, degrees of freedom, etc... are not reported.

7. the photomicrographs in Figure 4 are too small to see anything.  

Comments on the Quality of English Language

minor corrections needed

Reviewer 2 Report

Comments and Suggestions for Authors

In this manuscript, the authors claimed that early life stress may lead to visceral hypersensitivity via enhanced mast-cell activation and upregulation of histamine receptor expression in paraventricular CRH neurons. The data presented does offer support for this hypothesis; however, there are certain aspects that warrant further clarification. The paper is clearly written with minor (although numerous) grammatical errors.

Comments:

1. Were the separated pups kept warm during NMS? This is a critical question since at such a young age temperature change may be just as strong a stressor as the maternal separation. If temperature was controlled, it needs to be described with sufficient detail for replicability.

2. With the manual scoring of visceral pain threshold, were the quantitative measures like heart rate and EMG recorded? Was the scoring conducted blind and replicated / certified by an independent score?

3. How were cromolyn and histamine prepared for microinfusion?

4. In Figure 1C, while toluidine blue is recognized for its mast cell staining, its affinity for acidic entities such as DNA and RNA cannot be ignored. Due to the limited resolution of the presented image, discerning the presence of mast cell granules is challenging. Please annotate or mark the mast cells, particularly the activated mast cells in the figure. Ideally, the authors would also provide higher magnification images and detail how mast cells were identified.

5. In Figures 3A and B, was there a noticeable increase in c-fos activation and CRH protein expression following saline injection? What was the post-injection timeframe for the mice's sacrifice? Were the counts of total c-fos positive cells compared across conditions? 

6. In Figure 5, Western blotting depicted a decrease in H2R within the PVN, while immunohistochemistry demonstrated its colocalization with CRH staining. Are there other cell types within the PVN that also express H2R? If other cell types do express H2R, how might one ascertain that the observed upregulation of H2R specifically originates from CRH neurons?

7. In Figure 6, there appear to be disparities in phosphorylated (typically abbreviated as “phosho-PKA”, rather than “phosphor-PKA”) PKA and CREB levels between groups treated with NMS and ranitidine. Were there any variations observed in the total PKA and total CREB levels across these groups?

8. While the effect of NMS on mast cell and CRH neurons is well described, the underlying mechanism of how CRH neurons affect visceral hypersensitivity seems obscure. It would be beneficial if the authors could provide a more comprehensive discussion based on the existing literature regarding potential mechanisms.

9. For the p-values to be interpretable, the employed statistical tests must be named after each comparison presented. For example, how is the time course data compared in Figure 1A, Figure 2C, Figure 4C and Figure 5C? Are the authors using repeated measures comparisons? How are the comparisons made in Figure 3A & B and Figure 6 (these should be a one-way ANOVA across all data with multiple comparisons correction)?

10. There are numerous grammatical errors in the manuscript (too many to list). These are not major, and do not distract from understanding the otherwise well-written paper but it would still be recommended to have someone proof-read and correct the text.

Comments on the Quality of English Language

See comment #10.

Reviewer 3 Report

Comments and Suggestions for Authors

Authors should provide information about how significant their results in the abstract section briefly.

Authors should explain rationale behind selection of mice strain.

Authors should check their primary and secondary antibody information-some are not matching

Is seperation of 6 hours is not much? Where authors house animals in this period? If it is not much and coming from their previous knowledge authors should state and explain better

How pressure during colorectal baloon procedure quantified? Which rationale authors used to reach certain pressure? Did they directly go through 20-40 or step by step? If they reach directly are there any errors/overblowing?

Due to the histamine's and cromoyln's strong reactions, authors should explain dose selection or optimization of administration. Which kind of microinfusion pumps were used?

All material methods section should be detailed.

All figures should be prepared uniform

Results section should provide mathematical results also

Authors should avoid doing comments or discussing result in the result section. They should only report results.

Phopshor- in figure 6 should be revised.

discussion section should include how their results will help to future studies and what will outcomes will provide to any research field.

Round 2

Reviewer 1 Report

Comments and Suggestions for Authors

the revised version is fine. 

Reviewer 3 Report

Comments and Suggestions for Authors

Authors addressed the questions in a scientific manner and made needed changes to improve manuscript quality. It is suitable in that from for publication.